# HAL-X: Scalable hierarchical clustering for rapid and tunable single-cell analysis

**James Anibal**[1], **Alexandre G. Day**[2], **Erol Bahadiroglu**[1], **Liam O'Neil**[3¤a], **Long Phan**[1], **Alec Peltekian**[1], **Amir Erez**[1¤b], **Mariana Kaplan**[3], **Grégoire Altan-Bonnet**[1]*, **Pankaj Mehta**[2]*

**1** Immunodynamics section, Laboratory of Integrative Cancer Immunology, National Cancer Institute, Bethesda, Maryland, United States of America, **2** Department of Physics, Boston University, Boston, Massachussets, United States of America, **3** Systemic Autoimmunity Branch, National Institute of Arthritis and Musculoskeletal and Skin Diseases, Bethesda, Maryland, United States of America

☙ These authors contributed equally to this work.
¤a Current address: Max Rady College of Medicine, University of Manitoba, Winnipeg, Manitoba, Canada
¤b Current address: Racah Institute of Physics, Hebrew University, Jerusalem, Israel
* gregoire.altan-bonnet@nih.gov (GA-B); pankajm@bu.edu (PM)

**Data Availability Statement:** We make HAL-x publicly available at https://pypi.org/project/hal-x/. The raw data from the Lupus dataset can be accessed at https://doi.org/10.5281/zenodo.6332934.

## Abstract

Data clustering plays a significant role in biomedical sciences, particularly in single-cell data analysis. Researchers use clustering algorithms to group individual cells into populations that can be evaluated across different levels of disease progression, drug response, and other clinical statuses. In many cases, multiple sets of clusters must be generated to assess varying levels of cluster specificity. For example, there are many subtypes of leukocytes (e.g. T cells), whose individual preponderance and phenotype must be assessed for statistical/functional significance. In this report, we introduce a novel hierarchical density clustering algorithm (HAL-x) that uses supervised linkage methods to build a cluster hierarchy on raw single-cell data. With this new approach, HAL-x can quickly predict multiple sets of labels for immense datasets, achieving a considerable improvement in computational efficiency on large datasets compared to existing methods. We also show that cell clusters generated by HAL-x yield near-perfect F1-scores when classifying different clinical statuses based on single-cell profiles. Our hierarchical density clustering algorithm achieves high accuracy in single cell classification in a scalable, tunable and rapid manner.

## Author summary

Modern experimental techniques such as mass cytometry (CyTOF) make it possible to quickly make high-dimensional measurements on upwards of tens of millions of cells with single-cell resolution. An important problem in biology is to use these measurements to group together similar cells to identify biologically meaningful cell types that can be used to study disease progression, drug responses, and other clinical outcomes. However, the size and complexity of experimental data sets makes this problem computationally and theoretically extremely difficult. Here, we present a new algorithm HAL-X that accurately and quickly identifies cell clusters from biological data. Importantly, our algorithm

**Funding:** PM and AGD were supported by Simons Investigator in the Mathematical Modelling of Living Systems Grant, and NIH Grant No. R35GM119461. This research was supported in part by the Intramural Research Program of the NIH (MK and GA-B research groups). This project was initiated with a seed grant to GA-B and PM from the Gordon and Betty Moore foundation and Research Corporation through the Scialog program. The funders had no role in study design, data collection and analysis, decision to publish, or preparation of the manuscript.

**Competing interests:** The authors have declared that no competing interests exist.

does not require large amounts of memory. This eliminates the need for specialized high-end computing resources, allowing biologists to quickly analyze their data using a standard laptop or desktop computer.

This is a *PLOS Computational Biology* Software paper.

## Introduction

Technological advances have made it possible to collect huge single–cell datasets with numerous features *e.g.* transcriptomic profiles (scRNAseq) and/or protein expression (Flow cytometry, CyTOF, CyTEK or CiteSEQ). The ability to cluster such large, high-dimensional datasets is important for a variety of data-intensive fields ranging from biology to data mining. For this reason, there is a crucial need for fast, tunable and scalable clustering algorithms that work well in a high-dimensional setting on limited computational resources and that are at the same time reliable and robust. Clustering large high-dimensional datasets presents several challenges. Clustering in high dimensions suffers from the "curse of dimensionality" [1, 2]. The number of parameters in model-based clustering methods explodes and it becomes difficult to obtain accurate density estimates, a crucial ingredient in density-based clustering algorithms. Since many features are irrelevant or noisy, it is hard to construct meaningful similarity measures [3]. In order to circumvent these problems, one often needs to perform dimensional reductions and/or construct large similarity graphs, but both of these methods are computationally prohibitive for large datasets. Another difficult component of clustering is using the data to automatically learn the number of ideal clusters. Finally, arguably the most controversial and subjective part of clustering is validating clustering assignments when no ground-truth labels are available.

In scientific research, clustering algorithms provide many advantages and are widely used. These algorithms are easy to use, readily explainable, and do not require any form of annotated data. Nonetheless, clustering does have a significant downside: the requirement of prior knowledge about the data. A common task in modern biology is to cluster individual cells into distinct cell populations based on measurements in a high-dimensional feature space (mRNA or protein levels). Here, we focus largely on mass cytometry data (CyTOF) which allow measurements of more than 30 features, often for tens of millions of cells in a single sample [4, 5]. When performing exploratory analysis of unsupervised data, researchers often do not know the specificity (*i.e.* depth of clustering) needed to derive biologically-meaningful populations. A rare yet important population of cells may be lost when there are insufficient clusters. Conversely, it may be beneficial to study larger, well-understood populations that yield immediately actionable insights with robust statistical features.

To perform this type of comprehensive analysis, multiple sets of clusters must be generated across a broad spectrum of specificity. For currently available methods (i.e., Parc, Phenograph, FlowSOM), this requires training multiple models with different parameters to control the number of clusters [5–7]. In this scenario, time and/or memory complexity of these algorithms generate a significant problem, particularly in the area of cluster prediction. In the case of training, the solution is to perform dimensional reduction of the feature space (e.g. PCA the features) as well as possibly downsampling the data [5, 8].

In the context of single-cell analysis, there are a number of disadvantages to downsampling the dataset. While it is straightforward to identify generalizable cell populations with

downsampled data, downsampling makes it difficult to identify nuanced perturbations within populations. In addition, statistically significant biological features can also be lost. For example, biologists often try to identify biomarkers by training a supervised model for separating healthy cells from diseased cells. However, if the training data set is too small (i.e. there is too much downsampling), the trained model will exhibit severe overfitting because the downsampled data is not representative of true biological distributions.

One potential solution is to this problem is to train a clustering model on a downsampled dataset and use this model to predict labels on new data not seen by the model. Unfortunately, many of the most commonly used algorithms used for identifying cell clusters are not amenable to this approach because they are transductive methods: the underlying model changes when considering a new data point. As such, Parc and Phenograph lack the ability to predict labels for newly acquired data points. HDBSCAN [9], allows for label prediction after the model has been trained by estimating the position of the new point within a fixed hierarchical tree. To do this, the feature space of the test data must be reduced to match the training data. However, dimensionality reduction on a $25.10^6$x40 matrix (a typical dataset acquired by mass cytometry [5]) will overwhelm the computing resources available to most biologists. Furthermore, prediction becomes more computationally expensive as the amount of training data is increased. This is because labeling is performed by nearest neighbour search rather than inputting the data into a learned function.

In this paper, we introduce a hierarchical density clustering algorithm (HAL-x) that uses supervised linkage methods to overcome these challenges. Our algorithm builds upon the idea that clustering can be viewed as a supervised learning problem where the goal is to predict the "true class labels" from data [10–12]. Unlike these earlier works that rely on the ideas of cluster stability, we operationalize this concept by training an expressive supervised learning model (*i. e.*, support vector machines, random forests) to evaluate potential cluster assignments and by using out-of-sample performance as a metric of clustering goodness. By training a sufficiently expressive supervised model, we can ensure that the predictive power is limited by the reproducibility of our clustering assignments and not by the choice of classifier [13, 14]. Finally, we emphasize that using *out–of–sample* performance of expressive classifiers for clustering can be thought of as a generalization of density clustering to high-dimensional space, where direct density estimation is extremely hard. We empirically demonstrate that out-of-sample performance of a classifier trained on cluster labels is strongly correlated with the goodness of the clustering in the over-clustering regime. We leverage this relationship to generate robust clusters which generalizes well and enables the analysis of large biological datasets.

The output of our algorithm is a predictive classifier that encodes our clustering assignments. Our work makes 3 primary contributions to unsupervised learning:

1. Since we can easily predict labels with supervised classifiers, we can extensively down-sample our data when performing computationally expensive tasks such as dimensionality reduction and the construction of similarity matrices (tasks which scale as $O(n^2)$ or $O(n \log n)$).

2. Through the use of inductive learning methods, we can predict new data points at a minimum computational cost by simply applying the trained classifier on the full dataset. Moreover, we train these classifiers directly on unreduced data (i.e. without a dimensional reduction step), removing the need for expensive reduction of the test data.

3. Our model stores the classifiers corresponding to different out-of-sample error values. Thus, we can obtain multiple sets of cluster labels with a single model. We have converted our model into a simple python package such that users can easily generate multiple

clustering, simply by specifying the desired value for the minimum out-of-sample error (intuitively, more subtle clusters lead to high error values).

HAL-x removes the need for pre-processing the entire dataset or training a model on the entire dataset. Moreover, HAL-x allows rapid, scalable prediction of new points. Finally, a single trained HAL-x model can generate multiple clusterings at varied depths to account for the specificity/sensitivity trade-off. Our results show that supervised linkage methods have great potential in analyzing high-dimensional data with limited computational resources that are typically available to experimental biologists (i.e., 8 core laptop computers).

## Related work

Clustering has been extensively studied in the statistics and machine learning literature (see [3, 15] for full review). Prominent methods for clustering include density-based methods [16, 17], spectral clustering methods [18], and model-based clustering [2]. In the high-dimensional context, several algorithms have been developed that seek to rely on clustering on a subspace of features [19]. Other methods rely on dimensional reduction methods such as Principal Component Analysis (PCA) and Local Discrimination Analysis (LDA) in conjunction with K-mean clustering (often in an iterative manner) [20–22]. There has also been considerable performance on extending methods that generalize manifold learning and spectral clustering to a high-dimensional setting [23, 24]. Several algorithms have been proposed for clustering in high-dimensions by first projecting the data using non-linear embeddings using such as t-SNE and then performing clustering in the low-dimensional space [25]. A drawback of these approaches is that low-dimensional representations can compress the data in such a way that spurious clusters may form in the low-dimensional representation [26], making it extremely challenging to identify true clusters from artifacts of visualizations. Additionally, there exists specialized methods developed specifically for high-dimensional biological data such as Phenograph [5], PARC [7], FlowSOM [6] (see [27] for a comparison of some of these methods). However, these generally tend to require extensive computational resources to scale to large datasets (see analysis below).

More recently, state-of-the-art results were obtained by the DEC algorithm which combines deep embeddings with k-means clustering on the compressed, encoded representation [28]. The encoder weights and the means of the k-clusters were fine–tuned using a clever stochastic gradient descent–based training procedure. A particularly attractive aspect of this algorithm is that the cluster labels for new points can be calculated quickly by using the trained encoder to map points to the embedding space and finding the closest cluster center. However, an important drawback of this approach, shared by all k-means based approaches, is that it requires choosing the number of clusters by hand. This is often difficult to determine ahead of time for complex, high-dimensional data and retraining the deep encoder for different k can be time-consuming. Here we show how one can combine powerful non-linear embedding techniques, density clustering, and supervised methods that enforce self-consistency of cluster labeling to build a fast, accurate high-dimensional clustering algorithm that scales well to large datasets.

## Out-of-sample error captures clustering performance

Consider the problem of clustering n points $\{x_i \in X\}_{n_i=1}$ that live in a d-dimensional feature space. A (hard) clustering $C(x)$ maps each point in the feature space to a set of discrete labels in a set $K$, with the cardinality of $K$ equal to the number of possible cluster labels $|K| = k$. For example for k-means clustering, given a set of k-centroids $\{\mu_j\}_{k_j=1}$, the cluster labels are determined by labelling each data point by the closest cluster centroid. Empirical and theoretical

arguments suggest that meaningful clusterings $C(x)$ should not differ much when learned using different subsets of the data and different measure of stability have been proposed to quantify this idea [10–12, 29]. Here, we extend this idea to argue that good clustering $C(x)$ are maps that should generalize well. That is, given a clustering assignment generated on a subset of the data (a training data set), one should be able to accurately train a classifier to predict the cluster labels on unseen data (a test set). An important caveat is that the classifier we train should be expressive enough so that the predictive power of a classifier is limited by the properties of the clustering map rather than the classifier. It was recently shown that large, neural networks can achieve zero training error on random labels [13], suggesting that it is always possible to find such a classifier even for complex data. In practice, we have found that any reasonably powerful supervised learning algorithm (kernel-based SVMs, Random Forests, small neural networks) are sufficiently powerful to assess the goodness of clustering.

The intuition for using generalizability as a criteria is straightforward: clustering labels should reflect the underlying structure of the probability distribution from which the data is generated and we should be able to train an expressive classifier to learn this structure.

In order to understand why out-of-sample accuracy of expressive classifiers is a good proxy-metric for clustering accuracy, we must first nuance two distinct clustering regimes: the overclustering and underclustering regime:

**Overclustering (Fig 1a).** In the overclustering regime the number of predicted clusters is greater or equal than the number of true clusters. In such a situation, a classifier trained on clustering labels will generate decision boundaries across regions of relatively high density. Generating a decision boundary in a high density region increases the probability of overfitting and as such *reduces* the out-of-sample accuracy. This is precisely why out-of-sample accuracy is a good proxy for goodness of clustering.

**Underclustering (Fig 1b).** In the underclustering regime the number of predicted clusters is smaller than the number of true clusters. This is a regime where expressive classifiers will

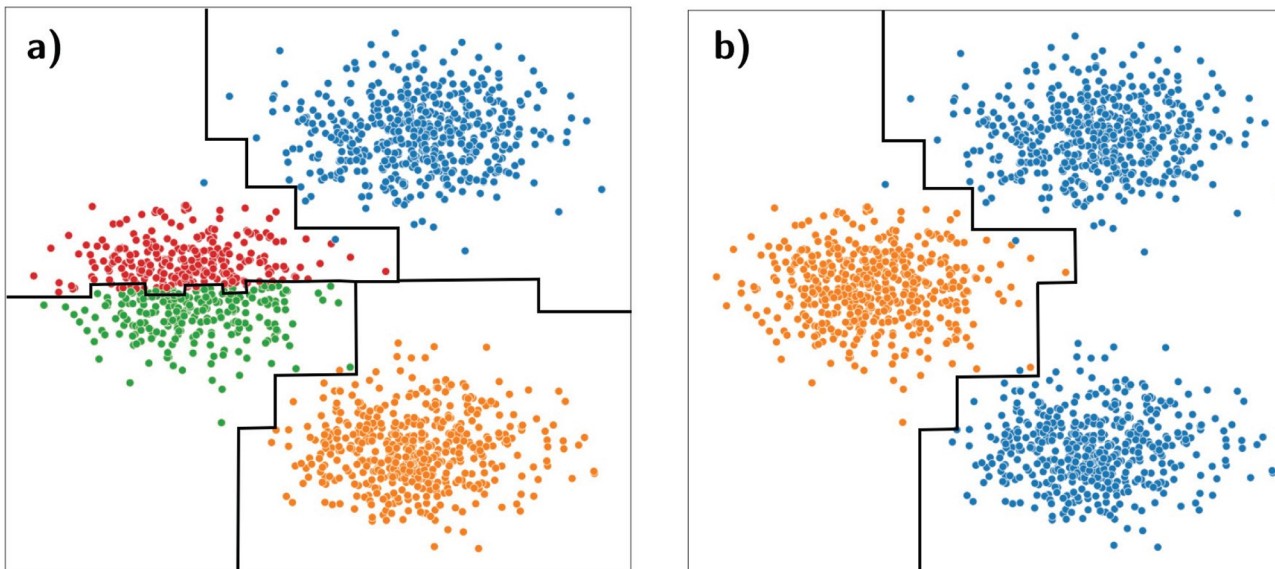

**Fig 1. Overclustering ($n_{Cluster}$ = 4) a) versus underclustering ($n_{Cluster}$ = 2) b).** The ground truth has 3 distinct clusters. The black lines represent the decision boundaries obtained via a random forest classifier trained on the clustering labels. a) Using over clustered labels for fitting a classifier leads to overfitted decision boundaries that separate regions of high density (red ang green region separation). b) For underclustered labels, the classifier groups together two clusters separated by a region of low density.

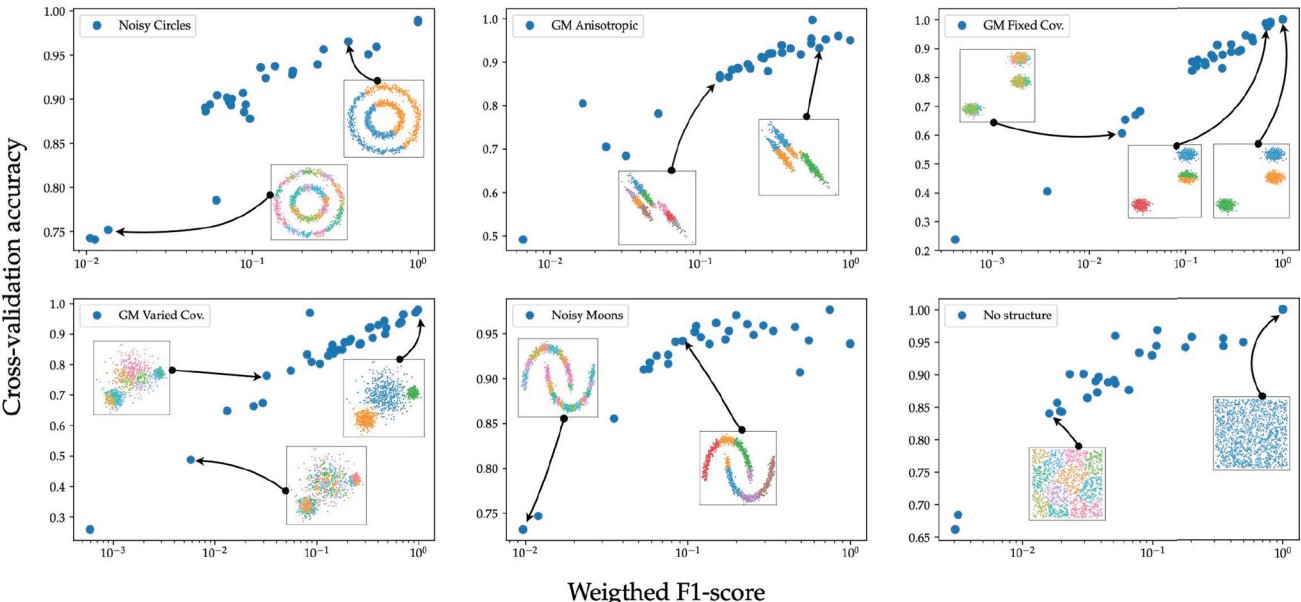

**Fig 2. Out-of-sample accuracy vs. weighted *F*-score (comparison to ground-truth) for various clustering algorithms.** We used out-of-the-box clustering algorithms *DBSCAN*, *spectral clustering*, *K*-means and *Meanshift* with various hyper-parameters and clustered 6 different benchmark two-dimensional datasets: Noisy Circles, Gaussian mixtures with different covariance structures, non-convex clusters (Noisy Moons) and a random noise map (No Structure). These datasets were taken from scikit-learn clustering methods as benchmark datasets (see S1 Methods for more details). The out-of-sample accuracy is computed by training a random forest classifier with 100 estimators on the clustering labels provided by the clustering algorithms. We use a 80/20 train/test split to train the classifiers and evaluate the out-of-sample error. Notice that the F-score and out-of-sample accuracy as measured are monotonically related.

indicate good performance but will not necessarily correlate with the goodness of clustering with respect to the ground truth.

In Fig 2, we empirically tested the relationship between out-of-sample accuracy and the goodness of clustering in the overclustering regime. We use 6 benchmark datasets from scikit-learn [30] and for each dataset generate multiple clustering assignments by tuning the hyper-parameters of various out-of-the box clustering methods from scikit-learn (see S1 Methods). For each generated clusterings, we train a random forest classifier and evaluate its accuracy on an hold-out test set. The out-of-sample accuracy shows a strong monotonic relationship with the weighted F-score. Moreover, we find a pearson correlation coefficient between the weighted F-score and the logarithm of the out-of-sample accuracy is 0.85 across the 225 sampled combination of datasets and clusterings.

We started by considering simple low-dimensional datasets designed to highlight the caveats and strengths of standard clustering methods [see scikit-learn]. For each dataset, we performed clustering using different clustering methods (DBSCAN, Spectral clustering, k-means, Meanshift) with varying parameters (see S1 Methods for the parameters used). The results are presented in Fig 2, where we plotted the out-of-sample accuracy of the trained classifier against the comparison to the ground-truth labels. The latter is quantified by using the weighted F1-score which uses the Hungarian algorithm for the optimal cluster matching and equally weights small or large clusters. As can be seen in the figure, clusterings with high F1-scores also tend to have good out–of–sample accuracy confirming that out-of-sample accuracy is a reasonable proxy for good clustering.

## Design and implementation

HAL-x has four major components that allow fast clustering of very large, high-dimensional datasets (Fig 3). First, HAL-x applies the t-SNE algorithm (or another dimensional reduction procedure such as UMAP) to reduce the dimensions on a down-sampled portion of the data. Second, HAL-x uses an approximate nearest neighbors algorithm and kernel density estimation to identify a initial set of "pure clusters" in regions above a specified density threshold. Third, HAL-x defines an extended density neighborhood for each pure cluster, identifying spurious clusters that are representative of the same density maxima and should be merged. Fourth, HAL-x builds a sparsely connected K-nearest neighbor (KNN) graph that connects each cluster with the *k* most similar clusters. The edges of the graph are labeled with the accuracy of a supervised classifier (i.e. SVM, random forest) that is trained to separate unreduced data points taken from the two clusters in the original high-dimensional feature space. The clusters with the lowest edge-score (lowest classification accuracy) are then merged. An overview of the basic workflow is shown in Fig 3. Runtime analysis and a pseudocode representation of the HAL-x algorithm can be found in S1 Methods.

### Dimensionality reduction

HAL-x is designed to cluster datasets with up to 100 million points embedded in a 50+ dimensional space (typical datasets collected in mass cytometry measurements). To do this, the data must first be projected into a low-dimensional plane. HAL-x uses the z-score technique to normalize a down-sample of the data. The size of the down-sampling must be adjusted depending on the dimensionality and quality of the dataset. For real experimental CyTOF datasets like those considered below, we have empirically found that using a down-sampled dataset consisting of one to two hundred thousand cells works well. This down-sampling is used to create a low-dimensional embedding, typically UMAP [31] or Fast Fourier Transform-accelerated Interpolation-based t-stochastic neighbor embedding (fitSNE) [32]. t-SNE is especially well-suited for density clustering methods since it preserves local ordination in the data, while repelling points that are far away.

A drawback of using a low-dimensional embedding such as tSNE or UMAP is that one must settle on a particular metric (usually Euclidean) to compute the initial similarity matrix before learning the embedding. This is problematic for data with highly correlated features and that have underlying labels invariant under global transformations (e.g. for images: translations and rotations). In order to alleviate this problem and generalize our approach to a broad class of high-dimensional datasets, we can perform principal component analysis which is known to help identifying uncorrelated linear combinations of features [15].

### Identification of pure clusters

HAL-x identifies an initial set of "pure", high-density clusters that form the starting point for agglomerative clustering. This is done by performing density clustering in the low-dimensional embedding space using a variation of the algorithm proposed in [17]. In particular, cluster centers are identified by finding isolated high-density points, with nearby neighbor mapped to the closest cluster center.

To estimate local densities, HAL-x performs an approximate nearest neighbors search via the locality sensitive hashing (LSH) search algorithm with random projections. For each point, the number of neighbors is scaled linearly with the number of points in the down-sampled dataset.

Once HAL-x has identified the neighborhood of each point, a Gaussian kernel density estimate is used to determine if a point has the highest density in the neighborhood. A full density

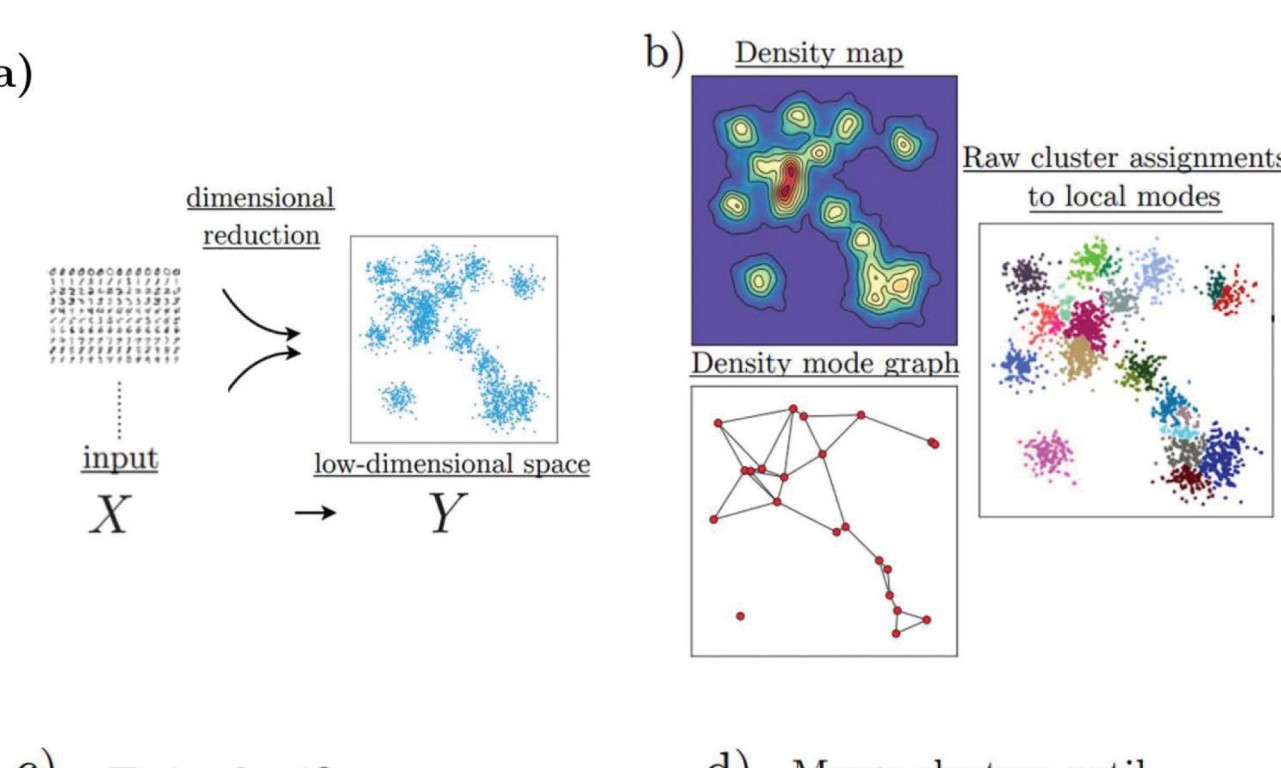

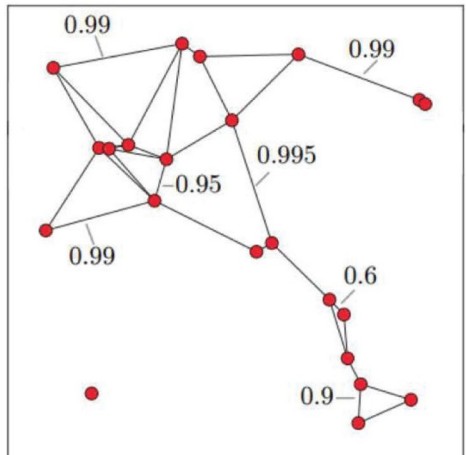

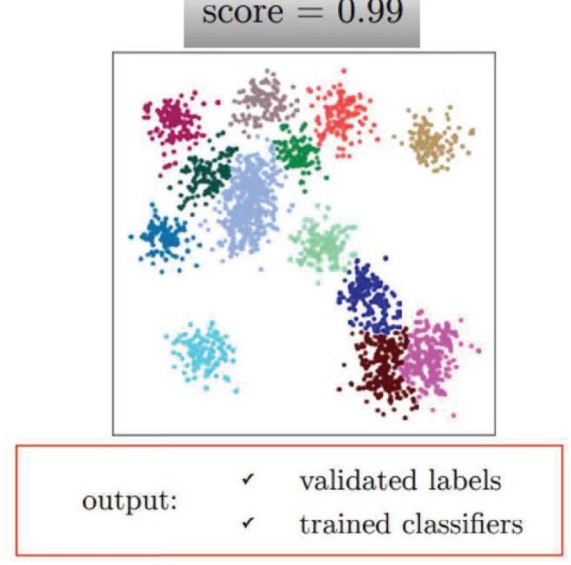

**Fig 3. Overview of the hierarchical agglomerative learning approach.** (a) For high-dimensional inputs, a dimensionality (e.g. UMAP [31], t-SNE, etc.) reduction step is required in order to obtain reliable density estimates. (b) In low-dimensional spaces, density maps can be easily computed. Initial clusters are selected to be the neighborhood of the density modes (see [17]). (c) A *k* nearest-neighbor graph is constructed by measuring similarity via the *out-of-sample accuracy* by training classifiers in the original high-dimensional feature space: each node represents an individual cluster and each edge has an associated weight given by the accuracy of the classifier. (d) Nodes are successively merged by pairs following the procedure until a desired out-of-sample accuracy is reached. The end result consists of an interpretable hierarchical classifier and robust clustering assignments. The classifier can be used to predict the labels of new data and potentially identify outliers.

map is constructed by assigning a Gaussian kernel of bandwidth $h$ to each data point ($x_i \in X$) and summing all kernels:

$$\hat{p}(x; \mathbf{X}, h) = \frac{1}{N} \sum_{x_i \in X} K_h(\mathbf{x}, \mathbf{x}_i), \tag{1}$$

where $K_h(\mathbf{x}, \mathbf{x}_i) = \exp(-(x - x_i)^2/h^2)/\sqrt{h}$. The bandwidth parameter can be thought of as a regularization parameter and sets how wide each kernel should be. We determine this parameter using a maximum likelihood estimation of Eq 1 over a testing set. We start by splitting the samples into a training and a testing set $X = (X_{\text{train}}, X_{\text{test}})$. Using the training set in order to construct and estimate $\hat{p}(x; \mathbf{X}, h)$, we can then compute the negative log-likelihood of the kernel density estimate over the testing set:

$$\mathcal{L}(X_{\text{test}}; X_{\text{train}}, h) = -\sum_{x_i \in X_{\text{test}}} \log \hat{p}(x; \mathbf{X}_{\text{train}}, h)$$

$$= -\frac{1}{N} \sum_{x_i \in X_{\text{test}}} \log \left[ \sum_{x_i \in X_{\text{train}}} K_h(\mathbf{x_j}, \mathbf{x}_i) \right]. \tag{2}$$

The bandwidth parameter $h$ can therefore be determined by minimizing Eq 2, which is a simple one-dimensional $f$ function of $h$.

After all centers have been identified, labels are assigned to each data point by simply comparing the relative density gradients between a given point and the cluster centers. After labels have been assigned to each point, if, for a given neighborhood, 99 percent of the cells share the same cluster center, that cluster is considered "pure" and is added to the initial set of pure clusters.

Once HAL-x identifies the pure clusters (i.e. performs the initial clustering), it performs a second, more rigorous validation of the point cluster assignment. We use two separate validation protocols to ensure that our initial neighborhoods have sufficiently high density and are not errant groupings that stem from a low density threshold. HAL-x uses the approximate nearest neighbors search algorithm a second time to define an extended neighborhood for each cluster center. This extended neighborhood includes points that are within a more relaxed density threshold than the one used for the approximate nearest neighbor search in the previous step. If there is a significant decrease in purity within the extended neighborhood, the conflicting clusters (representing the same density maxima) are merged to prevent unnecessary overlap.

## Supervised linkage

After the initial clustering (i.e. after the identification of the "pure" clusters), HAL-x performs agglomerative clustering by linking clusters that are hard to distinguish. This supervised linkage between clusters is performed in the original high-dimensional feature space. This is done by training supervised learning classifiers (commonly random forests or support vector machines) to distinguish all *pairs of clusters* from each other. In other words, if there are $N$ pure clusters, we train $N(N - 1)/2$ classifiers that distinguish between each pair of clusters. HAL-x then builds a minimally connected K-Nearest Neighbors (KNN) graph wherein each edge in the graph connected a cluster to the $K$ clusters that are the most difficult to distinguish from itself (i.e. to the $K$ clusters with the lowest out-of-sample accuracy) [33].

Using the KNN graph, HAL-x then performs a deeper sweep on each pair connected by an edge. This deeper sweep is performed with a larger ensemble of random forest classifiers built

on bootstrapped samples of the data, thereby reducing the risk of overfitting. When HAL-x has completed this deeper sweep, the two clusters with the lowest edge-score (classification accuracy) are merged. The KNN graph and the labels are updated accordingly. This process (starting with a deep sweep of all clusters connected by an edge) is repeated until there is only a single cluster that contains all the data points (corresponding a perfect cv-score of 1).

The classification ensemble that generated the final merge is stored in the root node of a tree data structure. We chose a tree data structure so that a path can be traced between any pure cluster and the root. Excluding the pure clusters, which act as the leaves, each cluster is a parent for the two child clusters that were merged to create this parent cluster. The classifiers used to separate the child clusters are stored in the parent node, and each edge (separating a parent and a child node) is linked to the accuracy value (cv-score) that led to the merge.

## Prediction

Using the tree data structure created during the training process, HAL-x can ultimately predict the multiple cluster labels for millions of unreduced data points. For this purpose, the user selects a minimum score (cv-score), which corresponds to the accuracy of the cross-validated supervised models. Every predicted label will correspond to a cluster that can be separated from all other clusters with a mean accuracy score greater than the cv-score value. A higher cv–score will result in fewer clusters, whereas a lower cv–score will result in a large number of clusters that are only slightly different from neighboring clusters (See Fig 4B and 4C). The default cv-score is 0.50: random classification. A cv-score of 0.50 will typically consider the full set of pure clusters when predicting the labels.

**Algorithm 1** HAL-x Prediction

```
Input: data matrix x, tree tree, float cvScore
parent = tree.root
for point in x do
  while True do
    if parent.edgeScore > cvScore then
      label = parent.ensemble.predict(point)
      parent = tree.getNode(label)
    else
      point.label = label
      break
    end if
  end while
end for
```

To make the predictions and thereby apply the clustering across all datasets, HAL-x first separates the data using the classifiers corresponding to the final merge that created the cluster containing all the data. Subsequent predictions are made using the classifiers from the children of the predicted cluster, further separating the data based on increasingly nuanced patterns. This process continues until the next predicted label would result in a cluster that is connected to the parent with an edge-score of less than cv-score. With the tree approach utilized here, a range of cv-score values can be efficiently checked for the same model, generating different sets of clusters depending on the level of population specificity desired by the user. Algorithm 1 highlights the simplicity of the HAL-x prediction technique via the cv-score parameter.

## Results

Here, we present the results of testing HAL-x on both synthetic and real-world datasets. First, we benchmarked HAL-x on the FlowCAP I dataset. We then generated multiple synthetic datasets containing Gaussian populations. We used these datasets to compare the scalability of

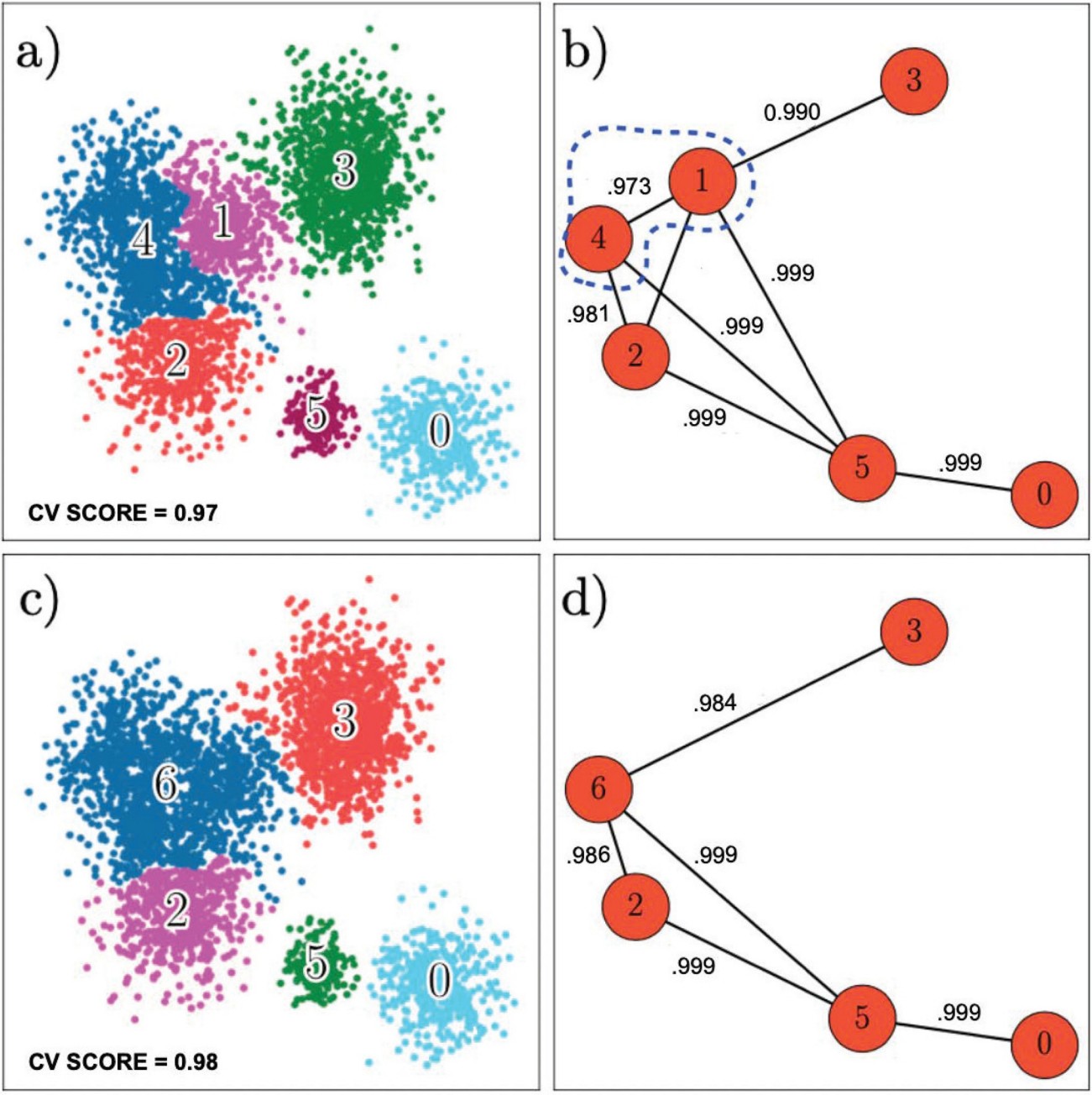

**Fig 4. Clustering based on out-of-sample error.** (a) Clusters generated with a cv-score of 0.97. (b) Graph illustrating out-of-sample error for each pair of clusters. There are no pairs of clusters with an out-of-sample error that is greater than 0.03 (accuracy value of 0.97). (c) Clusters generated with a cv-score of 0.98. Clusters 1 and 4 have been merged because the out-of-sample error was greater than 0.02. (d) Graph illustrating accuracy for each pair of clusters. There are no pairs of clusters with an out-of-sample error that is greater than 0.02 (accuracy value of 0.98).

HAL-x to Parc, Phenograph, FlowSOM, and HDBSCAN. We also tested HAL-x on a large, real-world mass cytometry dataset [4, 34], custom–generated at the NIH. This dataset contains 25.9 million single-cells taken from healthy individuals and from patients with Lupus [35]: these samples were profiled with an antibody panel quantifying 34 surface epitopes. After clustering, we performed supervised classification to determine if HAL-x clusters could be used to

**Table 1. F1 Measure of clustering accuracy for diverse methods, applied onto CyTOF datasets from [5].**

| Method | F1 for Levine-13 | F1 Levine-32 |
|---|---|---|
| HAL-x | 0.61 | 0.77 |
| Parc | 0.50 | 0.73 |
| Phenograph | 0.52 | 0.65 |
| HDBSCAN | 0.21 | 0.15 |

discriminate clinical statuses from leukocyte profiles: this provides a foundation for future experimental work and drug/biomarker discovery.

## Levine Benchmark datasets

We first tested HAL-x on two benchmark datasets from [5] to demonstrate the validity of HAL-x clusters: scalability is meaningless if the outputs are random. The first dataset is a flow cytometry dataset containing 81,747 human bone marrow mononuclear cells (BMMC) cells from 1 healthy tissue sample. These BMMC cells have 13 features corresponding to surface epitopes and a label corresponding to phenotype. The second dataset is a mass cytometry data-set containing 104,184 human bone marrow mononuclear cells (BMMC) cells from 2 healthy tissue samples. These BMMC cells have 32 features corresponding to surface epitopes and a label corresponding to phenotype. These phenotypes have been hand defined by flow cytome-try experts and represent our general understanding of leukocytes in bone marrow. We used HAL-x to cluster these single–cell datasets and compared the multiclass F1-scores with those achieved by three other clustering algorithms: Parc, Phenograph, and HDBSCAN (Table 1).

We see from Table 1 that HAL-x significantly outperforms Parc, Phenograph, and HDBSCAN on both these datasets, showing HAL-X is competitive with existing techniques in terms of accuracy.

## Synthetic datasets

HAL-x is designed as a convenient, accessible tool for biomedical researchers for this reason we wanted to assess how well HAL-x preformed on large datasets with limited computational resources. To do so, we generated a synthetic dataset with 10 million data points, the typical size of a dataset in single-cell biology, and tested HAL-x on an 8-core laptop computer—the typical computing resources available to a bench researcher. High-performance computing was not used in this study. Within this computing environment, we compare HAL-x to the three algorithms widely used for fast clustering of high-dimensional data: Parc, Phenograph, and HDBSCAN.

We generated datasets with $10^3$, $10^4$, $10^5$, $10^6$, and $10^7$ synthetic data points, each with 30 features. The size and dimensions of these synthetic datasets were chosen to match the struc-ture of mass cytometry data, which typically contains approximately 30 features (antibodies) and more than $10^7$ single cells. For HAL-x and HDBSCAN (algorithms which have prediction capabilities), we defined a 1 percent downsample to train on the datasets containing from $10^4$ to $10^6$ data points. For the dataset with $10^3$ points, we used a 10 percent downsample because 100 data points was insufficient for the perplexity of the fitSNE algorithm. We used a 0.5 per-cent downsample on the dataset with $10^7$ points so the memory required by the fitSNE algo-rithm would not exceed available resources in our computing environment. We do not perform any dimensionality reduction on the test dataset: without significant computing resources, this is typically infeasible for large datasets. Therefore, we are unable to reduce the

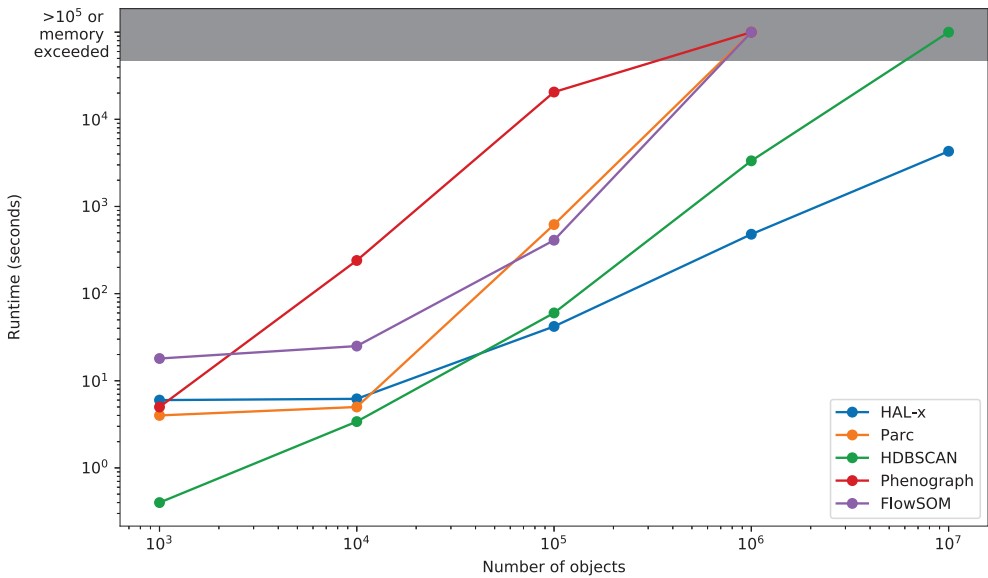

**Fig 5. Comparison of run times for training and label prediction on synthetic datasets of increasing sizes (comparing HAL-X with other clustering algorithms routinely used with cytometry data -Parc, FlowSOM, HDBScan & Phenograph).** The graybox indicates that the memory of laptop was exceeded for large datasets when using other algorithms than HAL-X.

training set for HDBSCAN, which requires both training and test data to have the same dimension. Parc, Phenograph, and FlowSOM cannot predict the cluster labels of new data points. For these methods, we attempted to learn the clustering model on the entire dataset (without downsampling). We used the default parameters for all the methods considered in this study.

In Fig 5, we show the scalability of HAL-x compared to 4 algorithms that are commonly used for single-cell data clustering. We see that all 5 algorithms are relatively equal at $10^3$ and $10^5$. At $10^5$, Phenograph is significantly slower than HAL-x, HDBSCAN,FlowSOM, and Parc. At $10^6$, Parc,FlowSOM, and Phenograph exceed the available memory within our computing environment. HDBSCAN can train a model on 500,000 cells and predict on $10^7$ cells, but the runtime is immensely slower than HAL-x. This observation is consistent with the predictive mechanism of HDBSCAN, which relies on nearest neighbour search rather than a set function learned by the supervised classifiers used in HAL-x.

## Lupus dataset

Finally, to illustrate the scalability and tunability of HAL-x, we tested HAL-x on a real-world mass cytometry datasets containing 25.9 million individual cells analyzed from 75 peripheral blood mononuclear cells (PBMC) samples and 75 polymorphonuclear (PMN) samples [36]. 49 of these samples were from Lupus patients and 26 of these samples were from healthy individuals. We designed a 38-marker mass cytometry panel with a focus on neutrophils as well as general phenotyping of blood leukocytes (see S1 Methods, S1 and S2 Figs for experimental details).

At the onset of this study, we assumed no prior knowledge regarding which populations whose differential frequencies would constitute good biomarkers for lupus and potential

targets for immunotherapy. Thus, we sought to generate 5 separate sets of clusters, each with a different level of population specificity, ranging from very rare to quite broad.

In under 1 hour, HAL-x generated 5 separate clusterings for the 25.9 million cells. Fig 6B and 6C presents the linear relationship between the cv-score parameter and the number of clusters, demonstrating the ability of HAL-x to widely vary the number of clusters using the same model.

For each HAL-x cluster (across our 5 different clustering depths), we trained a random forest classifier to separate healthy cells and Lupus cells. We discovered multiple clusters of PBMC cells and neutrophils wherein the healthy cells and lupus cells were classified with an AUC score of 0.82. The altered phenotypes of these cell populations (Fig 6E) can be considered novel biomarkers for Lupus. In particular, Cluster #42 for cv-score of 0.7 is a subpopulation of mature neutrophils (defined as $CD66ace^{+}CD66b^{+}$) whose levels of expression for CXCR1, CD15, CXCR2, IgA, CD66ace, CD66b, CD10, CD45RO, CD24 and CD16 best distinguishes between Lupus patients and Healthy donors (Fig 6D and 6E). These populations will be the subject of wet-lab experimentation seeking to highlight their functional relevance for Lupus pathology and to identify new molecular/cellular targets for lupus immunotherapies. Moreover, our results on the lupus dataset show the importance of varying the population specificity when searching for key populations in high-dimensional data. We observed here that several of the key cell populations were only identified when the cv-score was 0.7 or lower (deeper clustering). When the cv-score was higher (a more broad clustering), these important clusters are merged with other populations. This resulted in lower accuracy values and the loss of nuanced phenotypes which may have clinical significance. Our results show the importance of testing multiple clustering with varying depths. For example, if we had fixed the cv-score at 0.8, we would have missed several biomarkers and lost valuable information about the cellular landscape. HAL-x offered both the robustness, tunability and scalability to match the biological complexity of our Lupus dataset.

## Availability and future directions

We make HAL-x publicly available at: https://pypi.org/project/hal-x/.

The emergence of inexpensive platforms for large-scale single-cell analysis in biomedical applications is presenting new challenges to researchers attempting to interpret these immense, high-dimensional datasets. HAL-x integrates density clustering (reduced data) and supervised classification (unreduced data) to generate clusters that can be projected across tens of millions of single-cells.

The success of HAL-x on our lupus datasets shows the importance of clustering algorithms that: 1) can easily predict points for new data and 2) can interact with raw data. These are vital components in the time/memory efficiency achieved by HAL-x. Moreover, the simple HAL-x prediction algorithm (Algorithm 1) allows for one model to generate multiple sets of labels corresponding to varying levels of population specificity—a valuable advantage for biomedical researchers who need to cluster as deep as possible to identify key populations (biomarkers) of biological relevance and as shallow as possible to retain statistical significance in these biomarkers.

The success of HAL-x clustering when used for supervised classification tasks (*i.e.*identifying a novel cluster of neutrophils that distinguishes lupus patients from their healthy counterparts) suggests that our approach has a promising role in biomarker derivation and drug discovery. For example, the trends within specific subpopulations can be used to predict targets for new Lupus immunotherapies. More generally, the ability of HAL-x to generate a clustering model that can be propagated and tuned across multiple datasets will standardize

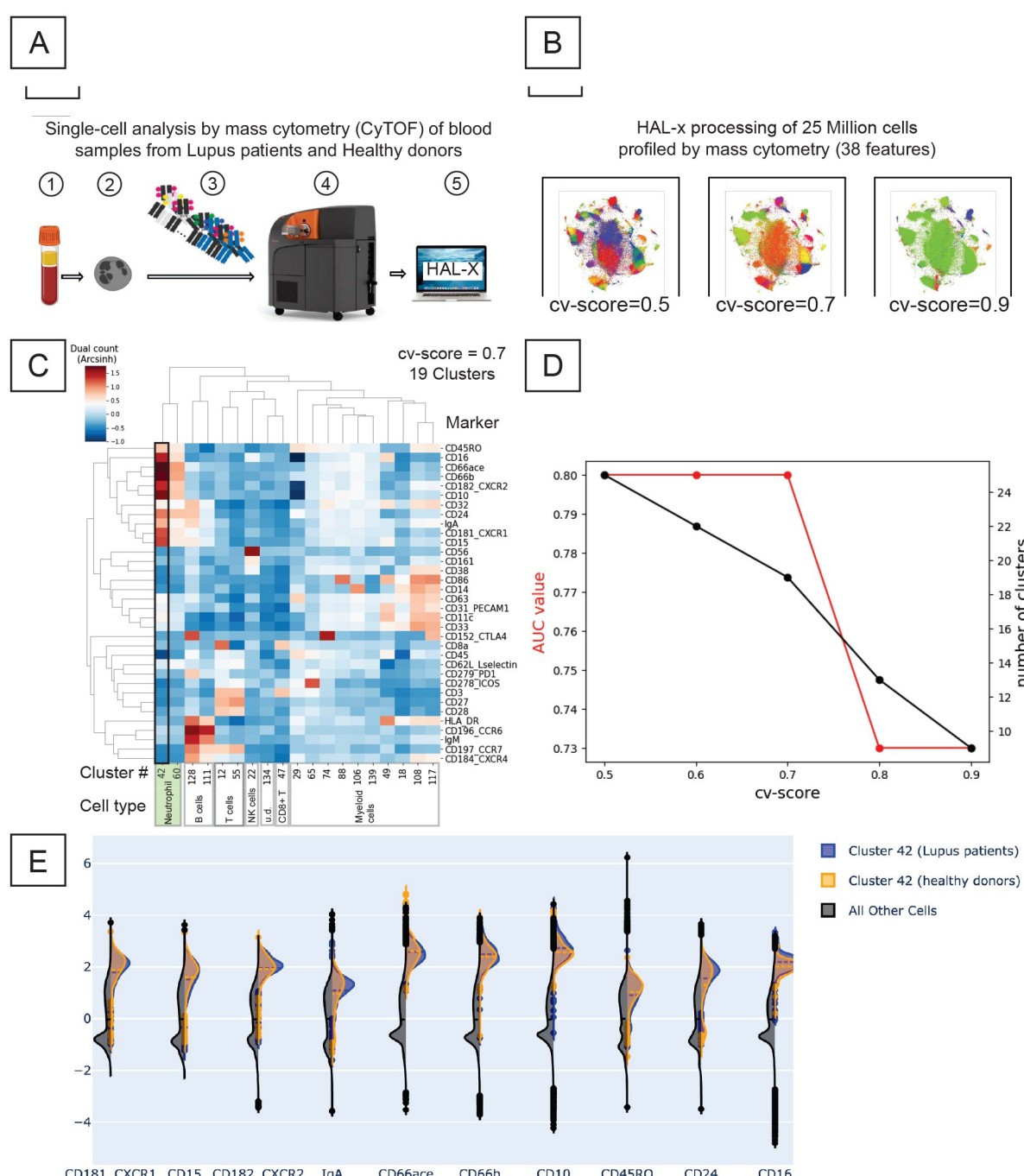

**Fig 6. Analysis of Lupus patients' blood profile using single-cell mass cytometry and HAL-x.** (A) The pipeline for profiling immune cells from Lupus patients and healthy donors, beginning with blood collection, continuing with mass cytometry, and culminating with HAL-x processing. (B) t-SNE visualizations of clusterings generated by HAL-x which correspond to various levels of population specificity (cv-score). (C) Heatmap of the average expression levels for the 19 leukocyte clusters defined by HAL-x for a cv-score of 0.7. The Cell type annotations were made hand. (D) A graph of AUC (*i.e.* Area under the Receiver Operating Curve (ROC)) for different cv-scores shows that the most important clusters with discriminatory power are not observed for high CV-scores. The best cluster identified by HAL-x (cluster #42 for cv-score of 0.7), separates individual neutrophils from Lupus and Healthy patients with an AUC of 0.82. (E) Markers on neutrophils (cluster #42) that can be used to distinguish blood samples from Lupus patients and from healthy donors. Note how subtle the difference is at the level of individual markers.

the ability of biological researchers to analyze vast amount of data and to communicate their results.

## Supporting information

**S1 Methods. Supplementary methods.**
(PDF)

**S1 Fig. Panel of metal-coupled antibodies used to profile the blood samples from Lupus patients and healthy donors.**
(PDF)

**S2 Fig. Raw .fcs file processing and gating strategy.** Left: gating for live cells (excluding Cisplatin-positive dead cells and EQ4 calibration beads. Right: gating for $CD45^+DNA(2n)$ singlet leukocytes.
(PDF)

**S3 Fig. Dynamical dashboard generated by HAL-x in order to inspect the different clusters and their hierarchy.** The dashboard has 3 panels. (left-side) The hierarchical relationship between the clusters. (top right side) The embedding map with the clustering labels. (bottom right) When choosing a cluster, this shows the profile of that clusters in terms of the original features.
(PDF)

**S4 Fig. Benchmark datasets used in figures of main text.** The red/blue/green coloring represents the ground-truth clustering labels when generating the datasets. The datasets are taken directly from scikit-learn clustering benchmark page. See https://scikit-learn.org/stable/modules/clustering.html for the exact definitions of the datasets.
(PDF)

**S5 Fig. HAL-x run for $N = 10^6$ synthetic data using a) t-SNE and b) UMAP for the primary embedding.** The results are very similar although we found that the UMAP approach ran approximately 50% faster in this instance.
(PDF)

## Author Contributions

**Conceptualization:** Alexandre G. Day, Amir Erez, Mariana Kaplan, Grégoire Altan-Bonnet.

**Data curation:** Erol Bahadiroglu, Alec Peltekian, Grégoire Altan-Bonnet.

**Formal analysis:** James Anibal, Pankaj Mehta.

**Funding acquisition:** Grégoire Altan-Bonnet, Pankaj Mehta.

**Investigation:** James Anibal, Alexandre G. Day, Liam O'Neil, Grégoire Altan-Bonnet, Pankaj Mehta.

**Methodology:** Alexandre G. Day, Mariana Kaplan, Pankaj Mehta.

**Resources:** Liam O'Neil.

**Software:** James Anibal, Alexandre G. Day, Long Phan, Alec Peltekian.

**Supervision:** Mariana Kaplan, Grégoire Altan-Bonnet, Pankaj Mehta.

**Validation:** James Anibal, Erol Bahadiroglu, Long Phan, Grégoire Altan-Bonnet.

**Visualization:** James Anibal, Grégoire Altan-Bonnet.

**Writing – original draft:** Grégoire Altan-Bonnet, Pankaj Mehta.

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
