## [Decision Letter · Decision Letter 0]

16 Dec 2021

Dear Dr. Altan-Bonnet,

Thank you very much for submitting your manuscript "Scalable hierarchical clustering with supervised linkage methods, for rapid and tunable single-cell analysis" for consideration at PLOS Computational Biology.

As with all papers reviewed by the journal, your manuscript was reviewed by members of the editorial board and by several independent reviewers. In light of the reviews (below this email), we would like to invite the resubmission of a significantly-revised version that takes into account the reviewers' comments.

Please ensure that the reviewers' comments are thoroughly addressed.

We cannot make any decision about publication until we have seen the revised manuscript and your response to the reviewers' comments. Your revised manuscript is also likely to be sent to reviewers for further evaluation.

Sincerely,

James R. Faeder

Associate Editor

PLOS Computational Biology

Manja Marz

Software Editor

PLOS Computational Biology

Reviewer's Responses to Questions

**Comments to the Authors:**

Reviewer #1: The manuscript by Anibal, Day, et al. introduces a new data clustering method (HAL-x) that is well-suited for studying large, high-dimensional data sets like those generated by mass cytometry. Data clustering is an important and timely topic given the emergence of large data sets in biology that can contain tens of millions of data points in a feature space with 40 or more dimensions. There have been a number of data-clustering efforts in this area, which the authors review in their paper. The authors overcome shortcomings of existing works by creating a workflow involving nonlinear dimensionality reduction, density clustering, and supervised linkage methods to create a cluster hierarchy on raw data. This work advances the field through the computational efficiency (scalability) of the method and the ability to readily generate multiple clusterings at varied depths.

Major concerns

1. In the “Related Work” section, the authors describe a shortcoming when using nonlinear embeddings like t-SNE and then performing clustering in the low-dimensional space: Namely, that the low-dimensional representations can compress the data and lead to spurious clusters. The first step of the HAL-x method uses t-SNE on a down-sampled portion of the data. It would be useful for the authors to explore how the initial embedding in low-dimensional space impacts the subsequent clustering by their algorithm.

2. The authors discuss (page 7) the possibility of preprocessing the input using an encoder “if necessary.” How can one decide if this step is necessary?

3. Regarding Figure 1, I was not convinced that “clusterings with high F1-scores also tend to have good out-of-sample accuracy confirming that out-of-sample accuracy is a reasonable proxy for good clustering.” Many clusterings with intermediate and even low weighted F-scores also have good out-of-sample accuracy. I encourage the authors to explain/explore this in more detail. I am not sure that I follow the logic/conclusion, or perhaps I am misunderstanding aspects of the figure.

4. The text in this section (“Out-of-sample error…”) refers to supplemental materials that I cannot find.

5. For the synthetic data set: How were the points distributed? How well did the algorithms perform on the data (i.e., how well did they cluster, assuming structured data was provided)? It seems like there is additionally opportunity here to assess the performance of the method.

6. I recognize that this is a methods-oriented paper. However, I think the paper would be strengthened by expanding the discussion of results for the Lupus dataset. For example, “if we had fixed the cv-score at 0.8, we would have missed several biomarkers and lost valuable information about the cellular landscape.” The authors don’t have to dramatically expand the scope of their paper, but this really leaves me hanging as a reader. I would like to know more about the biomarkers and/or what types of clusters are merged as cv-score increases.

Minor comments

- Some terms are not defined in Eqns 1 and 2.

- Page 9: What does “consider the full set of pure clusters” mean?

- Is FlowCAP I the same as the Levine benchmark datasets?

- Based on the wording, it was hard to tell if the lupus data was generated for this paper or was a previously published data set.

Typos

“tend to have out good of sample” (page 5)

“that are only slightly differences” (page 8)

“with 10 millions data” (page 11)

“constitute a good biomarkers” (page 12)

Reviewer #2: In this paper, Anibal et al. describe a novel method for data clustering, especially high-dimensional data such as that produced by CyTOF and similar experimental methods. Their method works by reducing the dimensionality of the data, developing a base set of clusters on this space, and then refining them. One of the key principles in this work is that a good clustering algorithm should produce clusters that generalize well to new data. The authors test their method against other existing approaches and find that their approach has advantages in both speed and accuracy.

This paper makes a valuable contribution to the literature on practical clustering methods. However, there are a number of points where the paper could be clarified or where important concepts are underexplored. In general, some details of the methods and simulations are unclear or have not been given, and the examples on real data could be explored further. My detailed comments are below:

1. I ran into several problems attempting to run the authors’ software, at least if this is run outside of Anaconda. In order to use the software, users must have the FFTW package installed because it is a requirement for fitsne. However, this is not currently mentioned in the installation notes, and a naïve installation through pip will fail without FFTW. The authors should note the dependency on FFTW and point to where it can be installled. The posted example also fails with the current version of sklearn due to changing conventions in the make_blobs function. I recommend that the authors change that line of the example to: “X,y = make_blobs(n_samples=10000,n_features=12,centers=10)” in order for the function call to complete correctly. The output in the example is also not easy to parse for a nonexpert. There is no output visualization, nor is there any apparent test quantifying goodness of fit (or some other metric) for how well the algorithm is working. I’m having trouble debugging this fully, but it looks like the visualization fails because at some point HAL calls for a plot using javascript, but this is run through a Python 2.7 call instead of Python 3 and the program fails to find the http.server module. I encountered these errors on a Mac with up to date versions of Python and all associated libraries.

2. One of the main points of emphasis in the paper is the ability to scale the sensitivity of the analysis to obtain more coarse or granular clusters (point 3 on page 3). The authors present real and simulated examples demonstrating this principle, but it would be helpful if some more detail was included, especially for the real data example. The simulated example presented in Figure 3 (note: no information is provided about the source of the data) gives a sense for how this works, but it appears that the definition of clusters can be quite sensitive to the chosen accuracy value. A small shift in the cv-score cutoff from 0.97 to 0.98 causes two large clusters to merge together. Of course, part of this may be due simply to the nature of thresholds, but this is not easy to see from the example. It would be helpful for future users of this method if an example were provided where a wide range of values for the accuracy cutoff is explored in a smooth way. The real data example provides something like this (Figure 5C), but it is difficult to appreciate how clustering changes as the threshold shifts given the large steps in cv-score values. It would also be helpful if the authors could present more detail on their results in the main text. At present, in this example, little data is presented beyond broad, qualitative statements (e.g., “several of the key cell populations were only identified when the cv-score was 0.7 or lower”; which populations are these, and how is it inferred that these populations are important?). Practically, it would also be helpful if such a worked example was provided together with the code, so that users of the algorithm will be well-prepared to perform this analysis themselves.

3. In the section on tests on synthetic data, the authors mention the size of the data set that they consider, but nothing is stated about how the data is structured. This is in principle an important choice, especially given that this section gives a comparison between different methods on the data. The authors should describe this in detail, and ideally the test data should also be available online (either as part of the repository for the code, in a database such as Zenodo, or as part of the supplementary material for this paper).

4. On page 7, the authors state that “we can also preprocess the input using an encoder if necessary” in place of t-SNE. At present, this statement is very vague, and it is not clear when the authors use t-SNE and when they use an encoder. How would the authors suggest that users of their software decide which approach to take for a particular data set? Can the consequences of different choices be explored on an example data set? How does this choice affect the run time?

5. Similarly, the authors use multiple classifiers for different data sets. In the examples in Figure 1, the authors use support vector machines, while for the Lupus data set they use random forests. Why? On page 4, the authors state that “any reasonably powerful supervised learning algorithm” can be used, but again this statement is rather vague and leaves users to wonder whether different algorithms may be appropriate for different circumstances, or whether these approaches are truly interchangeable.

6. The authors claim that the accuracy of predicting clustering for out of sample data is a good proxy for what constitutes good clustering, and I agree with their intuitive rationale. However, this is a bit difficult to see in Figure 1 because there are quite a few clusterings with high out of sample accuracy and low F-scores. It appears that this might be due to plotting multiple different types of data sets on the same plot, but this is still somewhat difficult to see. How well-correlated are the out of sample accuracy and F-score for each type of data? Also, in the materials that I have available, the data sets themselves are not described. It would be very helpful if this information was included in the supplementary information. In particular, I am somewhat confused by the “no_structure” data set – how does one evaluate clustering accuracy on a data set that has no intrinsic structure?

7. Despite its frequent use as a metric of clustering quality in the paper, the (weighted) F1-score is never explicitly defined.

Minor comments:

1. On page 4, the authors write: “…there exist[s] specialized methods developed specifically for high-dimensional biological data such as Phenograph, others, etc.” Here, Phenograph is the only method that is named. Are there other methods that should be listed here?

2. In Figures 4 and 5 many of the fonts are small and difficult to read. The clustering plots in Figure 5B are so small that it is very difficult to examine them in detail.

3. Given that the HAL-x software can produce visualizations, it would be helpful for users if some of these visualizations were included in plots in the paper. This requirement may already be satisfied – as explained above, I was not able to get plots to work with the software, and so I was not able to examine the output directly.

4. It seems that equation 2 should be described as the negative log-likelihood (or minus the log-likelihood) instead of simply the log-likelihood.

Reviewer #3: In this paper, the authors proposed HAL-x, a new hierarchical density clustering algorithm that can be scalable to a large data. HAL-x is especially designed for flow cytometry data, a large single cell data that often suffers from the scalability problem. While HAL-x has multiple interesting components, there are still multiple components that are not sufficiently clarified. I provide my comments in detail below.

1. The degree of down-sampling seems to be data-dependent. However, this is not a trivial task. For example, in the synthetic data analysis, the authors use very different values for different datasets, e.g., 10% vs. 0.5%. Can the authors provide some guideline to determine the degree of down-sampling?

2. fitSNE is proposed to be used for the dimension reduction purpose. Recently UMAP became more popular compared to t-SNE-based approaches. Can the authors provide justification to use fitSNE rather than UMAP?

3. It is proposed to map the data onto a two-dimensional plane using fitSNE. While 2D is great for the visualization purpose, it might not be optimal for the clustering purpose. Can the authors provide justification of using 2D or consider to use other dimensions?

4. In Equation 1, what is the mathematical meaning of arrow (->)? Convergence?

5. In the kernel density estimation, kernel and bandwidth are two critical choices and they have significant impact on the output. In the case of kernel, while Gaussian kernel is popular, Epanechnikov kernel is more popular and mathematically considered to be optimal. Similarly, there are various bandwidth selectors, including plug-in methods, cross-validation methods, etc. For example, Sheather & Jones method is a popular choice. Can the authors provide justification of using Gaussian kernel and a cross-validated MLE?

6. In Prediction step, the cv-score cutoff is a critical tuning parameter, which determines the number of clusters. While I acknowledge that hierarchical clustering is the power of HAL-x, biologists often need to determine a specific number of clusters for downstream analyses. Can the authors provide any guideline to determine the number of clusters? Is the default value (0.5) recommended for the general purpose?

7. For flow cytometry data, FlowSOM is the most popular algorithm for cell clustering and multivariate t-mixture is also likely popular. However, these popular algorithms are not included in the benchmarking. Please consider to include them in the benchmarking.

8. I cannot find the Author Summary section. Please provide this section in the manuscript.

**Have the authors made all data and (if applicable) computational code underlying the findings in their manuscript fully available?**

Reviewer #1: **No: **The source code is available. However, details about synthetic data sets used in Figures 1 and 4 are missing.

Reviewer #2: **No: **Test data and code used to evaluate different methods does not appear to be available at present.

Reviewer #3: Yes

PLOS authors have the option to publish the peer review history of their article (what does this mean?). If published, this will include your full peer review and any attached files.

Reviewer #1: No

Reviewer #2: No

Reviewer #3: No
---

## [Decision Letter · Decision Letter 1]

2 Jul 2022

Dear Dr. Altan-Bonnet,

We are pleased to inform you that your manuscript 'HAL-X: scalable hierarchical clustering for rapid and tunable single-cell analysis' has been provisionally accepted for publication in PLOS Computational Biology.

Best regards,

James R. Faeder

Associate Editor

PLOS Computational Biology

James Faeder

Associate Editor

PLOS Computational Biology

Reviewer's Responses to Questions

**Comments to the Authors:**

Reviewer #1: The authors have addressed my concerns.

Reviewer #2: The authors have responded to my original comments, and I have no further questions.

Reviewer #3: I believe my previous comments were well addressed by the authors and this manuscript is acceptable to be published.

**Have the authors made all data and (if applicable) computational code underlying the findings in their manuscript fully available?**

Reviewer #1: Yes

Reviewer #2: Yes

Reviewer #3: Yes

PLOS authors have the option to publish the peer review history of their article (what does this mean?). If published, this will include your full peer review and any attached files.

Reviewer #1: No

Reviewer #2: No

Reviewer #3: No

---

## [Editor Report · Acceptance letter]

20 Sep 2022

PCOMPBIOL-D-21-01438R1 

HAL-X: scalable hierarchical clustering for rapid and tunable single-cell analysis

Dear Dr Altan-Bonnet,

I am pleased to inform you that your manuscript has been formally accepted for publication in PLOS Computational Biology. Your manuscript is now with our production department and you will be notified of the publication date in due course.

With kind regards,

Zsofi Zombor
